# Identification and Tissue Expression Profiles of Odorant Receptor Genes in the Green Peach Aphid *Myzus persicae*

**DOI:** 10.3390/insects13050398

**Published:** 2022-04-20

**Authors:** Jingtao Liu, Jiaoxin Xie, Adel Khashaveh, Jingjiang Zhou, Yongjun Zhang, Hui Dong, Bin Cong, Shaohua Gu

**Affiliations:** 1College of Plant Protection, Shenyang Agricultural University, Shenyang 110866, China; knighteva@163.com (J.L.); biocontrol@163.com (H.D.); 2State Key Laboratory for Biology of Plant Disease and Insect Pest, Institute of Plant Protection, China Academy of Agricultural Science, Beijing 100193, China; xiejiaoxin18@126.com (J.X.); akhashaveh@caas.cn (A.K.); yjzhang@ippcaas.cn (Y.Z.); 3State Key Laboratory of Green Pesticide and Agricultural Bioengineering, Guizhou University, Guiyang 550025, China; jjzhouchina@163.com; 4Department of Entomology, China Agricultural University, Beijing 100193, China

**Keywords:** *Myzus persicae*, insect olfaction, odorant receptor, expression profile, genomic structure

## Abstract

**Simple Summary:**

The green peach aphid is a pest that mainly endangers vegetables. It spreads a variety of plant viruses through pricking and sucking plant juice. At the same time, it also secretes honeydew, which causes mildew and pollution and leads to the loss of many crops. Olfaction plays a very important role in the aphids’ search for hosts, avoidance of natural enemies, mating and oviposition. However, the molecular mechanism of olfactory recognition in the green peach aphid is not clear. Odorant receptors (ORs) are a type of olfactory neuron membrane receptor expressed on insect antennae and they play a role in identifying odorants from the environment. We identified 33 OR genes from the genome and transcriptomes of the green peach aphid and analyzed their phylogenetic relationship and genomic structure. We further examined the expression patterns in different tissues by real-time quantitative PCR (RT-qPCR). This study enriches our understanding of the olfactory system of the green peach aphid and will help to better apply it to the integrated control of a variety of pests.

**Abstract:**

The green peach aphid *Myzus persicae* (Hemiptera: Aphididae) relies heavily on its olfactory system to locate plant hosts, find mates, and avoid parasitoids or predators. The insect odorant receptors (ORs) have been proven to play a critical role in the perception of odorants from the environment. In the present study, 33 odorant receptor candidate genes including the Orco gene were identified from the antennal, head, legs and body transcriptomes of *M. persicae*. Phylogenetic analysis of ORs from seven different orders of insect species suggests that ORs from different insect species are highly divergent and most ORs from the same species formed monophyletic groups. In addition, the aphid ORs were clustered into six different sub-clades in the same clade. Furthermore, the genomic structure of the OR genes also tends to be consistent, suggesting that ORs from the family Aphididae have a relatively close evolutionary relationship. Reads per kilobase per million (RPKM) and tissue expression profiles analyses revealed that 27 out of the 33 MperORs were uniquely or primarily expressed in the antennae, indicating their putative roles in chemoreception. This work provides a foundation to further investigate the molecular and ecological functions of MperORs in the aphid–aphid, aphid–plant and aphid–natural enemy interactions.

## 1. Introduction

The green peach aphid, *Myzus persicae* (Hemiptera: Aphididae), is one of the most economically important crop pests worldwide. This aphid has an exceptional ability to colonize more than 400 plant species from more than 40 families for parthenogenetic reproduction, and uses peach, its primary host, for sexual reproduction [1,2,3]. Indeed, the status of this species as a pest is enhanced by its global distribution, remarkable efficiency as a vector of more than 100 different plant viruses, and an extremely broad host range [4]. The wide distribution of *M. persicae* in the world is due to its extremely high population adaptability to the environment, its wide genetic variability, and broad phenotypic plasticity. In addition, *M. persicae* is a typical host-alternating aphid species and usually heteroecious and holocyclic, with the winter sexual phase spent on the branches of peach trees and the parthenogenetic (asexual) summer generations spent on a wide number of secondary herbaceous hosts [3], for which chemical communication plays a critical role for their fast transference and precise positioning among different host-plants [1,5]. The genome of *M. persicae* was sequenced in 2017 [6] in order to better study the pest at the molecular level.

Olfaction is an important sensory modality of chemical communication that allows insects to perceive semiochemicals in a complex environment [7,8]. On the antennae, a variety of sensilla are distributed, within which usually contain two or more olfactory sensory neurons (OSNs). Odorant receptors (ORs) are expressed on the dendritic membrane of OSNs for the identification of a variety of volatile chemicals, including inter- and intraspecies semiochemicals and host odorants [9,10]. Insect ORs are seven-transmembrane domain G-protein-coupled receptors (GPCRs) and have an inverted membrane topology with an intracellular N-terminus and an extracellular C-terminus compared with vertebrate ORs [11,12,13]. Insect ORs usually form heteromeric complexes with a well-conserved odorant receptor co-receptor (Orco), which forms an odorant-gated ion channel with tuning ORs [14,15,16].

The first members of the OR family were discovered two decades ago [17,18,19]. Since then, insect ORs have been identified in many species, including *Tribolium castaneum* [20], *Locusta migratoria* [21], *Drosophila melanogaster* [22], *Bombyx mori* [23], *Apolygus lucorum* [24], *Microplitis mediator* [25], and *Manduca sexta* [26], with a high degree of divergence. The number of ORs is considerably varied between species, for example, there are 259 ORs in *T. castaneum* [20], 110 ORs in *A. lucorum* [24] and 79 ORs in *Acyrthosiphon pisum* [27].

In the present study, we identified 33 ORs genes from the antennae, head, legs and body transcriptomes of *M. persicae* combined with the data from the *M. persicae* genome (https://bipaa.genouest.org/is/aphidbase/myzus_persicae/, accessed on 13 May 2020) [27], and phylogenetically compared them with other insect ORs. The expression profiles of the ORs among different tissues were investigated using real-time quantitative PCR (RT-qPCR). In addition, the genomic structure of *M. persicae* ORs was analyzed. This work provides basic information that could be useful for the functional clarification of these odorant genes at the molecular level.

## 2. Materials and Methods

### 2.1. Insect Rearing and Sample Collection

A *M. persicae* colony was collected from infested Chinese cabbage (*Brassica rapa* L. ssp. *pekinensis*) leaves at the Langfang Experimental Station of the Chinese Academy of Agricultural Sciences, Hebei Province, China and established in the laboratory under the following conditions: 18–24 °C, 65–75% relative humidity, and a 16 h:8 h light:dark photoperiod. The colony was reared on tobacco (*Nicotiana tabacum* L.) seedlings in environmental chambers. For the tissue expression profile studies, around 2000 apterous adult aphids were dissected on ice under anatomical microscope (Olympus Corporation, Tokyo, Japan). Four different tissues (antennae, head without antennae, legs and body without head and legs) were collected separately in 1.5 mL centrifuge tubes and immediately frozen in liquid nitrogen and stored at −80 °C until use. Samples of each tissue were collected three times in the same way as three biological replicates.

### 2.2. RNA Extraction, cDNA Library Construction and Illumina Sequencing

Each tissue sample (50 mg) was used for total RNA extraction using Trizol reagent (Life Technologies, Carlsbad, CA, USA) following the manufacturer’s protocol. The concentration of the total RNA samples was measured on a NanoDrop 2000 spectrophotometer (Thermo Fisher Scientific, Waltham, MA, USA ) and the RNA integrity was further confirmed using a 2100 Bioanalyzer (Agilent Technologies, CA, USA. ). The cDNA library of each sample was constructed using NEB Next Ultra RNA Library Prep Kit for Illumina (New England Biolabs, Ipswich, MA, USA) according to the manufacturer’s protocol. Briefly, messenger RNAs were isolated from 10 μg of total RNA of each tissue using oligo (dT) magnetic beads and fragmented into short nucleotides using the fragmentation buffer supplied with the kit at 94 °C for 5 min. Each cleaved mRNA was transcribed to first strand cDNA using random hexamer primer and M-MuLV Reverse Transcriptase (RnaseH). The second strand cDNA was subsequently synthesized using DNA polymerase I, dNTPs and RNase H. After the end-repair, dA-tailing and adaptor ligation, the products were amplified by PCR and purified using the QIA quick PCR purification kit (Qiagen, Valencia, CA, USA) to create the final sequencing library. The cDNA libraries were pair-end sequenced using a PE150 strategy on an Illumina Hiseq2500 platform (Illumina, San Diego, CA, USA).

### 2.3. Bioinformatics Analysis

The raw reads were trimmed with Trimmomatic software (version 0.32). Clean reads were de novo assembled into unigenes with the Trinity short read assembler (version 20121005) using default parameters [28]. To annotate the unigenes, we performed a BLASTx search against Nr and Nt databases at the National Center for Biotechnology Information (NCBI) with an *E*-value cut-off of 1.0 × 10^−5^ [29]. Gene names were assigned to each unigene based on the best BLASTx hit with the highest score value. We use the BWA-MEM alignment algorithm [30] and HTSEQ program (version 0.6.1) [31] to align the RNA reads with the assembly and count the read numbers mapped to each unigene. Reads per kilobase per million (RPKM) were calculated for the assembled transcripts based on their mapping data according to the formula published by Mortazavi [32]. Thus, the RPKM of each unigene was calculated based on the length of the gene and read count mapped to this gene.

### 2.4. Identification of M. persicae ORs by Bioinformatics

The previously described ORs sequences in the *M. persicae* clone G006 assembly v2 were downloaded from AphidBase at the BioInformatics Platform for Agroeco system Arthropods (https://bipaa.genouest.org/is/aphidbase/myzus_persicae/, accessed on 13 May 2020). Genomic scaffold sequences found by tblastn (*E*-value=e^−100^) were used to construct putative OR sequences manually using Sequencher v4.5 (Gene Codes, Inc., Ann Arbor, MI, USA) and refined using SplicePredictor (http://deepc2.psi.iastate.edu/cgi-bin/sp. Cgi/, accessed on 17 June 2021. All *M. persicae* ORs identified in this manner were in turn used in successive tblastn searches to identify other candidate sequences. The green peach aphid OR genes were named “*M. persicae* ORx” with a number consistent with the pea aphid ORs [27]. Fragments with high similarity with MperORs were filtered as redundant sequences. Sequences possessing only one or two recognizable exons and encoding proteins less than 200 amino acids were discarded. Because of high divergence and/or discontinuity among scaffolds, not all detected OR genes could be entirely annotated. In these cases, amino acid sequences shorter than 200 amino acids were discarded as probable gene fragments. All genes reported in this study are full length OR genes. The *M. persicae* ORs were further confirmed by their transmembrane-domain structures using TMHMM [33] server version 2.0 (http://www.cbs.dtu.dk/services/TMHMM, accessed on 18 June 2021. The nucleotide sequences of identified MperORs have been deposited in GenBank under the accession numbers (OM628748-OM628780).

### 2.5. Phylogenetic Analysis

Amino acid sequences of candidate *M. persicae* ORs were aligned by ClustalX 2.0 using default settings with the OR sequences from *T. castaneum* [20], *L. migratoria* [21], *D. melanogaster* [22], *Bombyx mori* [23], *A. lucorum* [24] and *M. mediator* [25] (Appendix A). Phylogenetic trees were constructed based on the maximum likelihood by RAXML version 8 [34] with LG substitution matrix selected by the PROTTEST 3 program [35]. Node support was assessed using a bootstrap procedure of 1000 replicates. The phylogenetic tree of 298 ORs from three different aphid species (*M. persicae*, *A. gossypii* and *A. pisum*), and two plant bug species (*A*. *lucorum* and *Adelphocoris lineolatus*) [36] (Figure 1) was constructed using the same approach. The OR names and sequences that were used in phylogenetic analyses are listed in the Appendix A. 

### 2.6. Verification of OR Sequences by Cloning and Sequencing

The open reading frames (ORFs) of each identified OR sequence was predicted by Open Reading Frame Finder (https://www.ncbi.nlm.nih.gov/orffinder/, accessed on 16 June 2020). Then gene-specific primers (Appendix A) were designed using Primer Premier 5.0 software to clone the ORF of each MperOR gene. The template cDNA was synthesized from antennal RNA using the Fast Quant RT Kit (TianGen, Beijing, China) according to the manual. PCR reactions were carried out with 200 ng antennal cDNAs with 0.5 units of Ex Taq DNA Polymerase (TaKaRa, Dalian, China). The PCR amplification conditions were set as 95 °C for 2 min, followed by 36 cycles of 94 °C for 45 s, 56 °C for 1 min, 72 °C for 2 min, and a final extension at 72 °C for 10 min. The PCR products were gel-purified and subcloned into the pGEM-T Easy vector (Promega, Madison, WI, USA), and the insert was sequenced using an ABI3730XL automated sequencer (Applied Biosystems, Carlsbad, CA, USA. ) with standard M13 primers.

### 2.7. Genomic Structure Analysis of M. persicae ORs

The genomic DNA sequences of *M. persicae* OR genes were extracted with BLASTn using the mRNA sequences of *M. persicae* ORs as ‘query’. The mRNA-to-genomic DNA alignment of each OR gene was analyzed using the Splign online program (http://www.aphidbase.com/aphidbase/downloads, accessed on 21 May 2021). The predicted structure of each *M. persicae* OR gene was reconstructed and recorded as described above.

### 2.8. Expression Profiles of M. persicae ORs in Different Tissues

RT-qPCR was performed to examine the expression of *M. persicae* ORs genes. The cDNA of antennae, heads, legs and the body parts were synthesized using the PrimeScript RT Reagent with gDNA Eraser (TaKaRa, Dalian, China). An equal amount of cDNA (150 ng) was used as the RT-qPCR templates. RT-qPCR was carried out on an ABI 7500 Real-Time PCR system (Applied Biosystems, Carlsbad, CA, USA). The *β-actin* (GenBank Acc. XM_022309797) and *GAPDH* (GenBank Acc. XM_022315441) genes were used as reference genes for normalizing the expression of target genes and correcting the sample-to-sample variations (Appendix A). The primers used for the RT-qPCR were designed using Beacon Designer 7.90 (PREMIER Biosoft International) (Appendix A). 

To confirm the uniformity of the amplification efficiencies of the target and reference genes, a pilot experiment was conducted to examine the variation of ΔCt (Ct, Target−Ct, *β–actin*/*GAPDH*) with template dilution. Briefly, we used five serial dilutions of cDNA from each sample. For each dilution, amplifications were performed in triplicate using primers for the target gene and *β-actin* or *GAPDH*. The mean Ct was determined for both target gene and *β-actin* or *GAPDH*, ΔCt (the difference between Ct of target gene and Ct of reference gene) was calculated, and log cDNA dilution vs. ΔCt was plotted. The LinRegPCR program (version 11.0) [37] was used to calculate the RT-qPCR amplification efficiencies of the target and reference genes (Appendix A).

Each RT-qPCR reaction was conducted in a 25 μL mixture containing 12.5 μL of SuperReal PreMix Plus (TianGen, Beijing, China), 0.75 μL of each primer (10 μm), 1 μL of sample cDNA (150 ng/μL), 0.5 μL of Rox Reference Dye and 9.5 μL of sterilized water. The RT-qPCR cycling parameters were 95 °C for 15 min, followed by 40 cycles of 95 °C for 10 s and 60 °C for 32 s. Then, the PCR products were heated to 95 °C for 15 s, cooled to 60 °C for 1 min, heated again to 95 °C for 30 s and cooled to 60 °C for 15 s to measure the melt curves. The negative control, which used only sterilized water, was included in each experiment. To ensure reproducibility, each RT-qPCR reaction for each sample was performed in three technical replicates and three biological replicates. The expression level of each *M. persicae* OR genes relative to *β-actin* and *GAPDH* were calculated by using the comparative 2^−ΔΔCt^ method [38]. The comparative analyses of each target gene among various tissues were determined using a one-way nested analysis of variance, followed by Tukey’s honest significance difference test using the SPSS statistics 18.0 software (SPSS Inc., Chicago, IL, USA).

## 3. Results

### 3.1. Identification of M. persicae ORs

We identified 33 putative OR genes (32 typical ORs and one atypical co-receptor Orco) from the transcriptome and genome of *M. persicae* (Table 1). All *M. persicae* ORs are full-length with open reading frames (ORFs) ranging from 1104 to 1626 bp and a mean length of 1253 bp. The completeness of the MperORs’ length was judged by amino acid alignment with ApisORs. All MperORs showed the presence of predicted multiple transmembrane domains with an inside N-terminus, as usually observed for insect ORs. The TMHMM prediction showed that 7 MperORs (MperORco, MperOR10, MperOR21, MperOR23, MperOR37, MperOR42 and MperOR47) had seven-transmembrane domains (Table 1). The average identity of MperOR-DemlOR pairs is 12%. However, comparing the MperORs with ORs from closely related species, *A. pisum* and *A. gossypii*, demonstrates a not-so divergent relationship as the average identity of MperOR-ApisOR pairs is 78% and MperOR-AgosOR pairs is 64%. Furthermore, MperORs are very closely related and have an amino acid identity of 43–96% similar to ApisORs (Table 1, Appendix A–S8).

### 3.2. Phylogenetic Analyses of MperORs

The phylogenetic tree was built using 606 OR sequences from seven different insect species (*M. persicae*, *A. lucorum*, *B. mori*, *D. melanogaster*, *L. migratoria*, *M. mediator* and *T. castaneum*) in six different orders including Hemiptera, Lepidoptera, Hymenoptera, Diptera and Coleoptera (Appendix A). As illustrated in the phylogenetic tree, ORs are highly divergent between species and most ORs from the same species formed monophyletic groups (Appendix A). In the phylogenetic tree, 25 MperORs (OR17, OR20–25, OR29, OR35–45, OR47, OR51, OR53, OR64, OR69 and OR78) were clustered in a species-specific subgroup. In addition, 4 MperORs (OR4, OR5, OR9 and OR10) were grouped with TcasOR187 and MmedOR53, and MperOR2 and MperOR3 were most closely related to the LmigORs superfamily. Analogously, MperOR67 was clustered with two AlucORs and most closely related to the MmedORs superfamily. The extremely conserved Orco subfamily in the seven species was clustered in one branch with clear orthologous relationships.

The phylogenetic tree of Hemiptera ORs was built using 298 OR sequences from three different aphid species (*M. persicae*, *A. gossypii* and *A. pisum*), and two plant bug species (*A**. lucorum* and *A**. lineolatus*) (Figure 1). All aphid ORs are clustered to six homologous clades (lineages), and clearly separated from ORs from plant bugs. Interestingly, the OR67s from the aphid and plant bugs are clustered into one same clade (Clade 2). The Orco sequences of the aphid and plant bugs are highly conserved and clustered into one clade (Clade 1) with a bootstrap value of 100 (Figure 1).

### 3.3. Genomic Structure of M. persicae ORs

The genomic structures and the splice site of the intron–exon junctions of MperOR genes were analyzed based on the *M. persicae* genome annotations and the GT–AG rules [39]. The results revealed that the size of the genomic sequences of MperOR genes ranges from 1.73 to 10.87 kb with an average length of 4.57 kb. Twenty-three OR genes (*MperOR4*, *MperOR**17*, *MperOR**20–23*, *MperOR**25*, *MperOR**29*, *MperOR**35–37*, *MperOR**40–45*, *MperOR**47*, *MperOR**51*, *MperOR**53*
*MperOR**64*, *MperOR*
*69**,* and *MperOR**78*) have 5 introns and 6 exons, and 6 OR genes (*MperOR5*, *MperOR**9*, *MperOR**10*, *MperOR**24*, *MperOR**38* and *MperOR**39*) have 4 introns and 5 exons. The remaining 4 OR genes (*MperOR2, MperOR3, MperOR67* and *MperORco*) have 1, 3, 6 and 8 introns and 2, 4, 7 and 9 exons, respectively (Table 2, Figure 2). The intron lengths of each *MperOR* are variable. The average intron length is 3.32 kb, the longest intron length is 9.7 kb for *MperOR4*, and the shortest is 0.54 kb for *MperOR39* (Table 2). 

We found that *MperORco* has 9 exons, 7 of which are relatively similar in size with a length of 100–241 bp. *MperOR67* has 7 exons. It is consistent with the clustering in the phylogenetic tree that *MperORco* and *MperOR67* are separately clustered. In addition, *MperOR2* and *MperOR3* have only 3 and 4 exons. *MperOR4,*
*MperOR5,*
*MperOR9* and *MperOR10* have 5–6 exons with relatively similar length. Again, it is consistent that *MperOR2* and *MperOR3* are clustered together in the same branch, and *MperOR4,*
*MperOR5,*
*MperOR9* and *MperOR10* are clustered together in another branch. The number and length of the remaining 25 *OR* exons are very conservative. However, for *MperOR44*, the length of the first exon is 709–781 bp, and the length of the fourth exon is 117–159 bp. The average length of the first, second, third and fourth exon is 742 bp, 84 bp, 100 bp and 123 bp, respectively. In addition, the length of the fifth and sixth exon of *MperOR69* is 426 bp and 102 bp, which is just opposite to *MperOR24*, *MperOR38* and *MperOR39.*

We also analyzed the genomic clustering of MperORs. The results indicated that the 28 *MperORs* are distributed in 33 different scaffolds (Table 1). *MperOR20*, *MperOR21* and *MperOR22* as well as *MperOR40* and *MperOR41* constitute two gene clusters on scaffold 27 and scaffold 268, respectively (Figure 3). The distances between *MperOR21* and *MperOR**22*, *MperOR2**0* and *MperOR22*, *MperOR40* and *MperOR**41* are 10.74 kb, 4.44 kb, and 6.09 kb, respectively (Figure 3).

### 3.4. Expression Profiles of M. perszicae ORs in Different Tissues

We explored the expression profiles of *MperORs* in different tissues using RT-qPCR. For diluted templates, the absolute values of the slope of all lines (log cDNA dilution vs. ΔCt) were <0.1, and the real PCR amplification efficiencies of the target and reference genes were more than 1.90 (calculated using the LinRegPCR program and listed in the Appendix A). Therefore, the efficiencies of the target and reference genes were similar in our analysis, and the 2^−ΔΔCt^ calculation method can be used for the relative quantification. 

The OR transcripts displayed tissue-specific expression patterns (Figure 4). The results showed that the transcripts of *MperOR2*, *MperOR3, MperOR4, MperOR5, MperOR9, MperOR10, MperOR20, MperOR21, MperOR22, MperOR23, MperOR25, MperOR29, MperOR35, MperOR36, MperOR37, MperOR38, MperOR39, MperOR40, MperOR41, MperOR43, MperOR51, MperOR53, MperOR64, MperOR67, MperOR69, MperOR78,* and MperORco were highly expressed in the antennae. In contrast, *MperOR42* and *MperOR44* were highly expressed in the head and the body, respectively. However, the remaining ORs showed a wide range of expression patterns. Among them, *MperOR45* was predominantly expressed in the body and the antennae, while *MperOR17* and *MperOR24* were mainly abundant in both the head and body.

Along with RT-qPCR measurements, the RPKM analysis also demonstrated the relatively high abundance of *MperORs* in the *M. persicae* transcriptome (Table 1). The RPKM and RT-qPCR results are in the analogical trend. The RPKM value showed that 27 ORs were highly expressed in the antennae, which was consistent with the trend of the RT-qPCR, although the specific values were slightly different. For example, the RPKM values of *MperOR17*, *MperOR24*, *MperOR42* and *MperOR45* indicated they are highly expressed in other parts of aphids, and consistent with the results of RT-qPCR. It should be noted that although the RPKM results of *MperOR44* and *MperOR47* and RT-qPCR results show they are highly expressed in the body, the RPKM values are not significant. 

## 4. Discussion

With the increase in insect genome and transcriptome sequencing projects, particularly in recent years, very large numbers of ORs have been identified in different insect species. In the present study, we profiled the transcriptomes in the antennal, head, body and legs of *M. persicae* using RNA-seq technology and annotated 33 full-length OR genes. Previous research revealed that OR genes are highly divergent in numbers and sequence among different insect species due to the requirement for each species to detect a panoply of odor signals that relate to their specific life histories. The number of *M. persicae* ORs identified here is less than the number of ORs identified in other sucking insects, for example, 79 in *A. pisum*, of which 48 OR genes encode apparently full-length, intact and potentially functional ORs [27]. Similarly, in *A. gossypii*, 45 ORs have been identified including 22 ORs full-length ORs [40]. Furthermore, MperORs are very closely related and similar to ApisORs, suggesting that aphid OR genes may be evolved from common ancestors without subsequent duplication. This suggests that they are probably homologous genes and may have similar function.

In the same species, different ORs play a role in recognizing different odorants. In the *M. persicae*, the average identity of 33 MperORs is 38%. On the other hand, compared with other insects such as *D. melanogaster*, OR genes of the *M. persicae* indeed display a high degree of divergence with DmelORs. This implies that ORs have great differences among species, although they may be homologous in functional classification. Specifically, most of the identified MperORs are highly expressed in the antennae, suggesting that they may be involved in the green peach aphid’s recognition of different host odors or pheromones. Furthermore, the number of MperORs identified here is less than the number of ORs identified in *D**. melanogaster* (58 *ORs*) [22], *B**. mori* (54 *ORs*) [23], *M**. mediator* (54 *ORs*) [25] and *T. castaneum* (177 *ORs*) [20], indicating the species uses a more narrow range of odors to detect hosts and/or conspecifics, or alternatively OR-mediated olfaction is less central to its life cycle, or that olfaction is less important than other senses. The phylogenetic analyses clearly divided most of aphid ORs into species-specific homology clades. However, OR67s of *M. persicae* and *A. pisum* are clustered with OR67s of *A. lucorum* and *A. lineolatus* with a bootstrap support of 100 (Figure 1), and with a relatively high protein identity (53%); a similar finding was also found in OBPs and CSPs between the aphid and plant bug [41], suggesting these olfactory genes may have a common function in these sucking insects.

By analyzing the genomic structure, we determined the number and length of introns and exons of 33 *MperORs*. The conserved genomic structure of 20 *MperORs* suggests that they have experienced recent gene duplication events, and the distribution of the clustering in the evolutionary tree analysis suggests that they may have similar functions.

To better understand the function of MperOR genes, tissue-specific expressions were evaluated by using RT-qPCR. MperOR genes exhibited diverse expression patterns, which could be briefly classified into five types, according to their level of expression: (1) antennae-enriched ORs, (2) head-enriched ORs, (3) body-enriched ORs, (4) head and body-enriched ORs, and (5) antennae and body-enriched ORs. It was also reported that some ORs could be expressed in a variety of tissues apart from the olfactory organs. Similarly, many ORs are also distributed in the head, especially near the mouthparts, thus OR42 could be possibly involved in the identification of odor substances at close distances [42]. As a co-receptor with ORs, *MperOrco* is highly expressed in the main olfactory organ—the antennae. The major alarm pheromone component (*E*)-β-farnesene (EβF) emitted from the cornicles of aphids [43] is recognized by large placoid sensillum neurons, which express ApisOR5 on the sixth antennal segment in *A. pisum* [44]. ApisOR5 and its orthologues in *M. persicae* (MperOR5) and *A. gossypii* (AgosOR5) show a high identity rate (77%), suggesting that their function is possibly conserved in aphids. Due to the parthenogenetic reproductive pattern of wingless aphids, the genes highly expressed in antennae, head and body may have been expressed before the birth of offspring aphids. This suggests that these genes may play a role in the embryonic development of aphids, or may have other unknown functions different from sensing smells. OR44 with a high body-specific expression may be related to the distribution of ORs at the end of the abdomen. The tissue expression profiles of each *MperOR* gene evaluated by the RPKM values are mostly consistent with the RT-qPCR results, but with several exceptions, this may be attributed to the different detection methods in transcriptome analysis and RT-qPCR. The RT-qPCR results can only represent the differential expression among the four tissue samples. The RPKM values, however, can compare all transcript expression levels among different tissues and each single transcript within one sample [45], for example, like most ORs, the RPKM values of all *MperORs* within four tissues showed most of them are enriched in the antennae. However, comparing the RPKM values in the antennae of all ORs can show that the expression depth of Orco in antennae is much higher than other OR genes, supporting the notion that Orco is widely expressed as OR-Orco complex [16].

## 5. Conclusions

In summary, we identified 33 candidate odorant receptors that may function in odorant perception in the green peach aphid, *M. persicae* from transcriptomes and genome data. By analyzing the phylogenetic relationships between other insects and the genomic structure of ORs, we speculated on the possible reasons for the evolution of OR in green peach aphid. As a crucial first step toward understanding their functions, a comprehensive examination of the expression patterns of these MperOR genes in different tissue samples was performed using RT-qPCR. We identified 27 MperOR genes that were highly expressed in antennae. The results of our study will provide a valuable foundation for further elucidating the mechanisms of olfaction in *M. persicae*, which could also help to use ORs as targets to regulate insect olfactory behavior and broaden the applications of available tools for effective control of pests.

## Figures and Tables

**Figure 1 insects-13-00398-f001:**
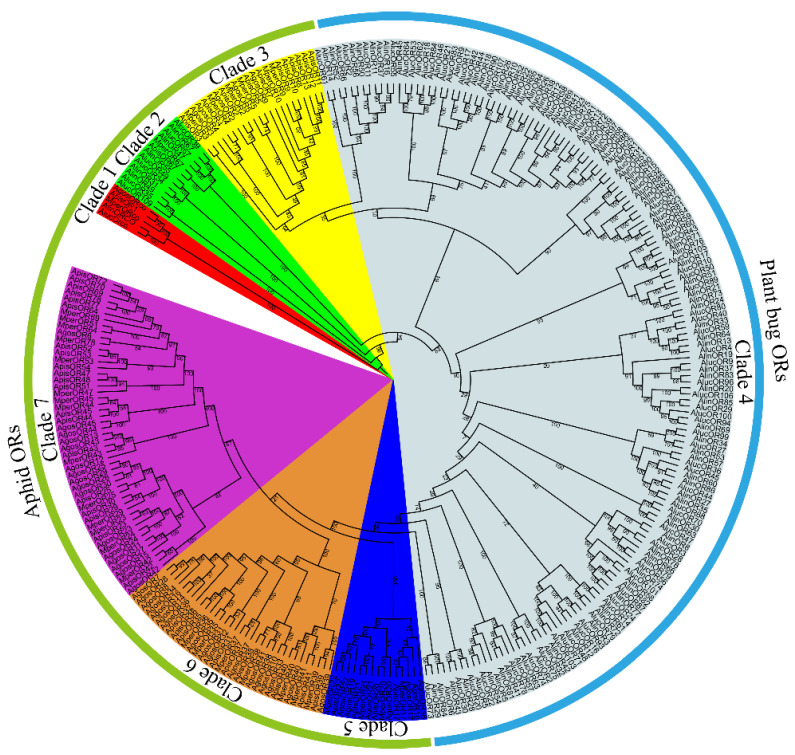
Phylogenetic tree of 298 ORs from three different aphid species (*M. persicae*, *A. gossypii* and *A. pisum*), and two plant bug species (*A**. lucorum* and *A**. lineolatus*). The protein names and sequences that were used in the phylogenetic analysis are listed in Appendix A.

**Figure 2 insects-13-00398-f002:**
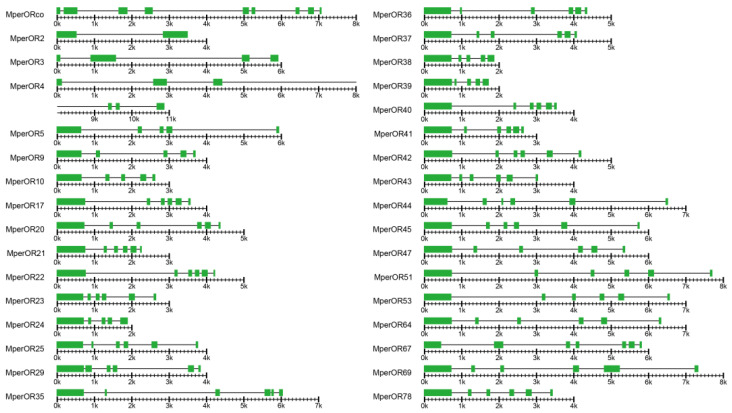
Genomic structure of *M**. persicae* OR genes. The green rectangles and hairlines between two green rectangles represent the exons and introns, respectively. The length has been shown to scale with a scale bar under each *M. persicae* OR gene; every minor mark represents 100 bp.

**Figure 3 insects-13-00398-f003:**
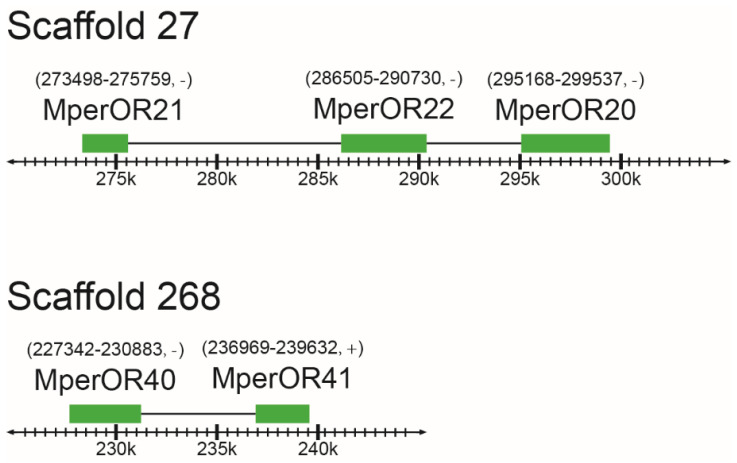
The tandem arrays of OR genes in *M**. persicae*. The genomic sequences of *M. persicae* were downloaded from the AphidBase at the BioInformatics Platform for Agroecosystem Arthropods (https://bipaa.genouest.org/is/, accessed on 13 May 2020).

**Figure 4 insects-13-00398-f004:**
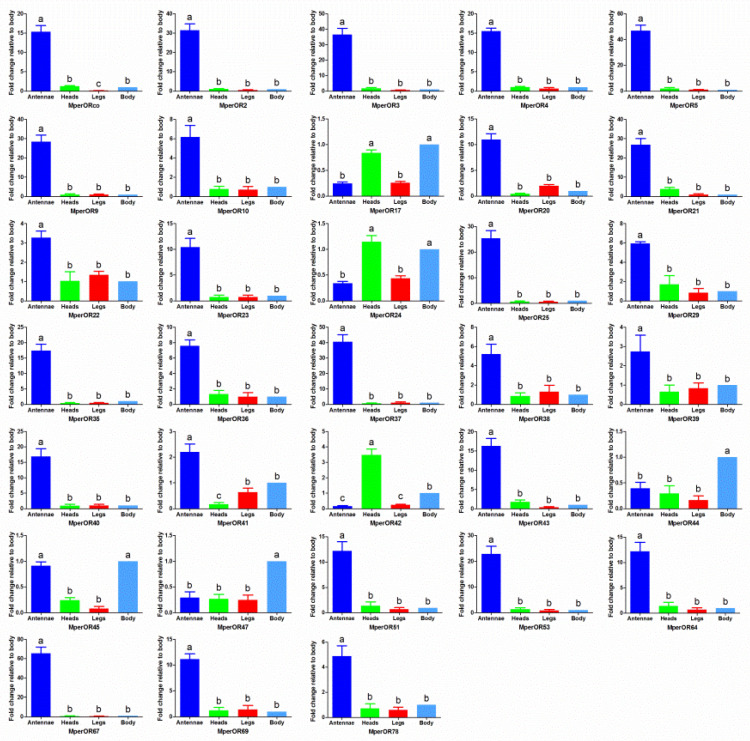
The OR transcript levels of *M. persicae* in different tissues assessed by RT-qPCR. The error bars present the standard error, and the different letters (a, b, c) indicate significant differences (*p* < 0.05). This figure was presented using *GAPDH* as the reference gene to normalize the target gene expression and correct sample-to-sample variation; similar results were also obtained with *β-actin* as the reference gene.

**Table 1 insects-13-00398-t001:** List of OR genes in *M. persicae*.

Gene Name	ORF (bp)	Amino Acids	Scaffold *	Antennae RPKM	Head RPKM	Legs RPKM	Body RPKM	Homology Search with *A. Pisum*		
								Most Similar	%Identity	TM Domains
MperORco	1494	498	173 (307,518–314,584)	16.04	1.32	0.15	1.05	ApisOR1	96	7
MperOR2	1203	401	200 (329,352–332,846)	4.19	0.14	0.07	0.14	ApisOR2	93	5
MperOR3	1206	402	27 (538,815–544,733)	2.76	0.14	0.04	0.08	ApisOR3	94	8
MperOR4	1167	389	2 (1,669,537–1,680,404)	0.66	0.25	0.08	0.12	ApisOR4	94	6
MperOR5	1104	368	185 (390,783–396,720)	1.28	0.08	0.00	0.02	ApisOR5	87	8
MperOR9	1107	369	52 (795,957–799,660)	1.23	0.03	0.03	0.03	ApisOR9	82	6
MperOR10	1122	374	448 (232,245–234,867)	1.23	0.03	0.03	0.03	ApisOR10	68	7
MperOR17	1296	432	54 (655,522–659,088)	1.36	4.15	1.44	5.60	ApisOR17	84	6
MperOR20	1266	422	27 (295,168–299,537)	3.45	0.14	0.62	0.32	ApisOR20	89	6
MperOR21	1272	424	27 (273,498–275,759)	1.39	0.20	0.05	0.05	ApisOR21	84	7
MperOR22	1290	430	27 (286,505–290,732)	0.09	0.00	0.00	0.00	ApisOR22	80	6
MperOR23	1248	416	128 (606,788–609,435)	0.42	0.00	0.00	0.02	ApisOR23	90	7
MperOR24	1224	408	128 (619,838–621,712)	1.49	5.02	1.90	4.38	ApisOR23	43	4
MperOR25	1203	401	1011 (10,792–14,561)	1.30	0.01	0.01	0.05	ApisOR25	80	5
MperOR29	1242	414	495 (178,577–182,414)	0.21	0.06	0.00	0.01	ApisOR29	72	6
MperOR35	1221	407	139 (21,194–26,450)	1.32	0.01	0.01	0.08	ApisOR35	59	6
MperOR36	1221	407	0 (1,027,879–1,031,452)	0.19	0.02	0.00	0.00	ApisOR36	75	6
MperOR37	1251	417	533 (71,681–75,764)	1.95	0.01	0.05	0.04	ApisOR37	88	7
MperOR38	1221	407	16 (1,217,873–1,219,749)	0.26	0.03	0.07	0.05	ApisOR38	84	6
MperOR39	1188	396	257 (222,543–224,270)	0.12	0.00	0.02	0.03	ApisOR39	93	5
MperOR40	1266	422	268 (227,342–230,883)	0.53	0.00	0.01	0.00	ApisOR40	86	6
MperOR41	1248	416	268 (236,969–239,632)	0.61	0.04	0.18	0.28	ApisOR41	84	6
MperOR42	1290	430	129 (610,966–615,169)	0.12	2.27	0.21	0.65	ApisOR42	75	7
MperOR43	1254	418	138 (86,064–89,109)	1.32	0.15	0.01	0.08	ApisOR43	92	8
MperOR44	1140	380	115 (575,915–582,437)	0.10	0.05	0.00	0.18	ApisOR44	59	8
MperOR45	1296	432	13 (1,040,637–1,046,400)	0.50	0.22	0.05	0.64	ApisOR45	58	5
MperOR47	1302	434	109 (556,358–561,734)	0.08	0.00	0.05	0.13	ApisOR47	73	7
MperOR51	1293	431	1192 (10,158–17,868)	0.53	0.06	0.00	0.03	ApisOR70	70	6
MperOR53	1281	427	700 (104,988–111,553)	0.84	0.04	0.00	0.02	ApisOR53	70	6
MperOR64	1290	430	380 (259,518–265,853)	0.53	0.06	0.00	0.03	ApisOR64	71	5
MperOR67	1239	413	354 (145,937–151,752)	5.87	0.06	0.05	0.09	ApisOR67	81	6
MperOR69	1626	542	304 (223,678–231,005)	0.36	0.04	0.06	0.02	ApisOR69	59	5
MperOR78	1290	430	140 (453,957–457,387)	0.36	0.05	0.03	0.08	ApisOR78	62	6

* AphidBase scaffold ID (start…stop nt, orientation).

**Table 2 insects-13-00398-t002:** The introns and exons of OR genes in *M. persicae*.

Gene Name	Genomic DNA Size (bp) and Strand (+/−)	No. of Intron	Total Length of Introns (bp)	Average Intron Size (bp)	No. of Exon	Total Length of Exons (bp)	Average Exon Size (bp)
MperORco	7066(**+**)	8	5572	697	9	1494	166
MperOR2	3494(**−**)	1	2291	2291	2	1203	602
MperOR3	5918(**+**)	3	4712	1571	4	1206	302
MperOR4	10,867(+)	5	9700	1940	6	1167	195
MperOR5	5937(+)	4	4833	1208	5	1104	221
MperOR9	3703(+)	4	2596	649	5	1107	221
MperOR10	2622(+)	4	1500	375	5	1122	224
MperOR17	3566(**−**)	5	2270	454	6	1296	216
MperOR20	4369(**−**)	5	3103	621	6	1266	211
MperOR21	2261(**−**)	5	989	198	6	1272	212
MperOR22	4227(**−**)	5	2937	587	6	1290	215
MperOR23	2647(**−**)	5	1399	280	6	1248	208
MperOR24	1874(**−**)	4	650	163	5	1224	245
MperOR25	3769(+)	5	2566	513	6	1203	201
MperOR29	3837(+)	5	2595	519	6	1242	207
MperOR35	5256(**−**)	5	4035	807	6	1221	204
MperOR36	3573(**−**)	5	2352	470	6	1221	204
MperOR37	4083(+)	5	2832	566	6	1251	209
MperOR38	1876(+)	4	655	164	5	1221	244
MperOR39	1727(+)	4	539	135	5	1188	238
MperOR40	3541(**−**)	5	2275	455	6	1266	211
MperOR41	2663(+)	5	1415	283	6	1248	208
MperOR42	4203(+)	5	2913	583	6	1290	215
MperOR43	3045(+)	5	1791	358	6	1254	209
MperOR44	6522(**−**)	5	5382	1076	6	1140	190
MperOR45	5763(+)	5	4467	893	6	1296	216
MperOR47	5376(+)	5	4074	815	6	1302	217
MperOR51	7710(**−**)	5	6417	1283	6	1293	216
MperOR53	6565(+)	5	5284	1057	6	1281	214
MperOR64	6335(+)	5	5045	1009	6	1290	215
MperOR67	5815(**−**)	6	4576	763	7	1239	177
MperOR69	7327(+)	5	5701	1140	6	1626	271
MperOR78	3430(+)	5	2140	428	6	1290	215

## Data Availability

The data presented in this study are available in Appendix A.

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
