# Peer review of "Identification and Tissue Expression Profiles of Odorant Receptor Genes in the Green Peach Aphid Myzus persicae"

_insects, 2022, doi:10.3390/insects13050398_

Round 1

Reviewer 1 Report

The article related to ORs in M. persicae is sound and it deals with the most studied olfactory proteins to date. Identification is crucial for further analysis, and in that way, this study has merit. Bioinformatics are well describe and applied, as well as qPCR experiments. Discussion is needs a bit more work in terms of comparisons with other ORs, not limited to aphids. English editing is also necessary. I attach a pdf file with some minor comments and corrections in form. Overall, I believe this article can be published in Insects after authors address my revisions.

Author Response

Response to Reviewer 1 Comments

Point 1: L32: “play”.

Response 1: Sorry for this mistake, we have corrected it as “play”.

Point 2: Line 33: Orco.

Response 2: Sorry for this mistake, we have corrected it as “Orco”.

Point 3: Line 81: keep abbreviations, in this case ORs.

Response 3: Sorry for this mistake, we have changed odorant receptor to ORs.

Point 4: Lines 218-221: this part of the paragraph seems more related to materials and methods rather than a description of results.

Response 4: As the reviewer 1 suggested, we have removed this paragraph to Materials and methods 3.1.

Point 5: Lines 323-325: I suggest to include RPKMs in a heatmap and then, compare it with qRT-PCR results.

Response 5: The RPKM values have been added in the Table 1, so we do not think additional RPKM heatmap is necessary. We have compared the RT-qPCR results with the RPKMs in the discussion part in the revised manuscript.

Point 6: I suggest authors to ask for English editing, especially in discussion section.

Response 6: The revised manuscript especially the discussion section was re-edited by JJ Scientific Consultant Ltd., England, UK.

Point 7: Discussion is needs a bit more work in terms of comparisons with other ORs, not limited to aphids.

Response 7: Thanks for this valuable suggestion. In the revised manuscript, we have added the comparisons of aphid ORs with other insect ORs.

Point 8: Line 376: was EbF abbreviated before?

Response 8: We are sorry for this mistake. We have corrected the “EβF” as “(E)-β-farnesene (EβF)” in the revised manuscript.

Other grammar and spelling mistakes all have been checked and corrected in the revised manuscript.

Reviewer 2 Report

The authors have done a descent job to identify odorant receptors from peach aphid, M. persicae from transcriptomic and genomic analysis. They have compared ORs with different genus and with other aphid species. The data presented is informative and pays way to further understand species capacity to infest wide range of hosts. Having said that, I have few concerns listed below that authors could address.

L32: “play”

L94: spacing

L91 to 100: It appears to me that there were no biological replication included in the study (?)! Is this study based on one biological study?

L235-240: The phylogenetic analysis with 606 ORs from different species such as D. mel, A, lucorum, B.mori etc do not add anything as new finding, and it has been shown repeatedly that insect ORs are diverse within and between genus. Furthermore, the tree in unreadable with that many sequences, so it is not possible to verify, and no conclusions can be drawn from such phylogenetic analysis.

Why did authors did not include other chemosensory receptors in their study?

Author Response

Response to Reviewer 2 Comments

Point 1: L32: “play”.

Response 1: Sorry for this mistake, we have corrected it as “play”.

Point 2: L94: spacing.

Response 2: Sorry for this mistake, we have corrected it as “16 h : 8 h”.

Point 3: L91 to 100: It appears to me that there were no biological replication included in the study (?)! Is this study based on one biological study?

Response 3: Thanks for this professional comments. As mentioned in L206-207, three biological replicates were used in this study, in order to describe more clearly, we have add “Samples of each tissue were collected three times in the same way as three biological replicates.” in the last of this paragraph.

Point 4: L235-240: The phylogenetic analysis with 606 ORs from different species such as D. mel, A, lucorum, B.mori etc do not add anything as new finding, and it has been shown repeatedly that insect ORs are diverse within and between genus. Furthermore, the tree in unreadable with that many sequences, so it is not possible to verify, and no conclusions can be drawn from such phylogenetic analysis.

Response 4: Thanks for this professional comments. In the revised manuscript, we have removed this figure to Figure S3 and provided a much higher resolution for clearly reading. Furthermore, we have re-built the phylogenetic tree using 298 OR sequences from three different aphid species (M. persicae, A. gossypii and A. pisum), and two plant bug species (A. lucorum and A. lineolatus), and discussed the phylogenetic relationships between the aphids and plant bugs in the results and discussion parts

Point 5: L32: Why did authors did not include other chemosensory receptors in their study?

Response 4: Thanks for this valuable suggestion. Like ORs, other chemosensory receptors such as IRs and GRs, may also play essential roles in aphid olfaction, however, In this paper we just want to fucus on the ORs, other chemosensory proteins will be studied in our future work.

Other grammar and spelling mistakes all have been checked and corrected in the revised manuscript.
